# Unifying Pixel-Labeling Vision Tasks by Sequence Modeling

## Abstract

Developing a single neural network that can perform a wide range of tasks is an active area of research in computer vision. However, unifying models for pixel-labeling tasks presents significant challenges due to the diverse forms of outputs and the reliance on task-specific structures. In this paper, we propose UniPixTask, a model that unifies pixel-labeling tasks by modeling them with discrete vocabulary codes. UniPixTask consists of two main components: a Sequence Learner that produces a unified expression and a Dense Decoder that constructs latent codes for dense prediction outputs. This combination enables UniPixTask to model the complex task space using unified discrete representations while maintaining high-quality output for pixel-labeling tasks. We evaluate UniPixTask on three pixel-labeling tasks: semantic segmentation, surface normal estimation, and monocular depth estimation. The experimental results show that UniPixTask is a promising approach for unifying pixel-labeling tasks. Our code will be available.

## 1 Introduction

Researchers have been working on developing general models that use a single neural network to perform multiple tasks. A key aspect of achieving this goal is to build unified representations for multiple modalities and tasks. The rise of autoregressive sequence models parameterized by Transformers (Vaswani et al., 2017) has led to the use of discrete vocabulary codes as a popular method for achieving unification. Compared to continuous and high-dimensional representations, discrete representations have more potential to fit different modalities and tasks. Language can be represented as a sequence of symbols, which are inherently discrete. Similarly, visual tasks such as keypoints detection and object detection produce discrete sets of keypoints or bounding boxes. Therefore, the use of discrete output is a key direction for the unification of multitask and multimodal, avoiding the need to add an additional decoder head on top of the continuous output to perform the discrete representation tasks.

Recent models (Jaegle et al., 2021b;a; Wang et al., 2022; Alayrac et al., 2022) have made various attempts in this regard. For instance, Pix2Seq (Chen et al., 2021) quantizes the coordinates of bounding boxes and class labels of object detection into discrete tokens. Pix2Seq v2 (Chen et al., 2022a) further expresses the discrete forms of more tasks like instance segmentation and keypoint estimation. Unified-IO (Lu et al., 2022) targets a wider set of vision and language tasks and homogenizes them into a sequence of discrete tokens.

However, for dense prediction tasks which produce pixel-level predictions of images, unifying them still poses great challenges. Labels for these tasks have diverse forms and may be continuous values. Simply quantizing these outputs into discrete variables can lead to a loss of accuracy. Moreover, unlike most tasks that only require a global representation, pixel-labeling tasks focus on local semantic details, necessitating pixel-level outputs that are dependent on feature resolution. However, due to the quadratic increase in sequence length, the traditional model is difficult to scale.

To address these problems, in this paper, we propose UniPixTask, a novel unified approach for modeling dense prediction tasks using latent discrete tokens. Our model is depicted in Figure 1 and we contribute in three ways:

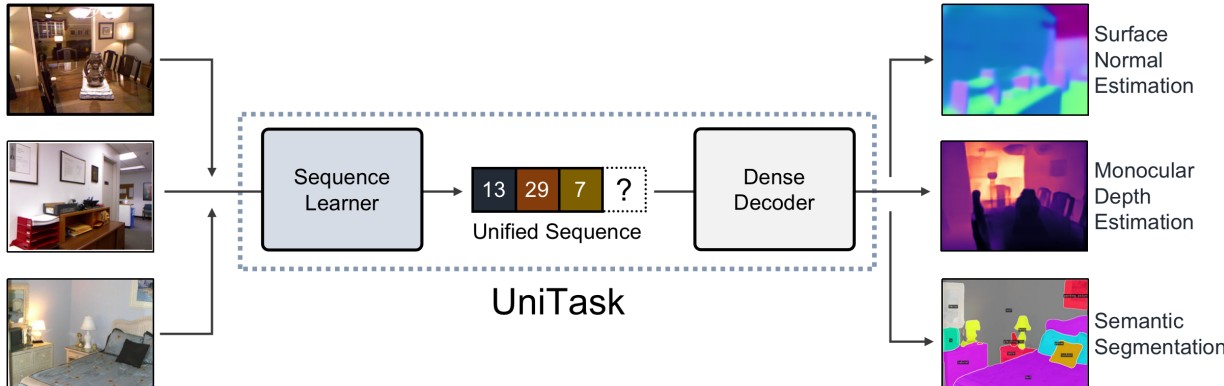

Figure 1: Overview of the UniPixTask framework.

- Sequence Learner and Dense Decoder work jointly to achieve discrete representation modeling for dense prediction tasks. Sequence Learner is efficient at predicting discrete vocabulary tokens while Dense Decoder is capable of generating task-specific outputs.

- Our compression and restoration modules enable efficient modeling of feature representations using a small number of latent codes. By restoring in the channel and spatial dimensions, we are able to efficiently reconstruct features from discrete codes with minimal information loss. This allows us to model complex pixel-labeling tasks while maintaining high-quality output.

- As a unified approach, UniPixTask is well suited to multiple tasks, including semantic segmentation, surface normal estimation, and monocular depth estimation. It achieves competitive results with well-established task-specific models.

## 2 Related Work

### 2.1 Pixel-labeling Vision Tasks

Pixel-labeling, also called dense prediction, refers to tasks that produce pixel-level outputs for images. These tasks perform pixel-level classification or regression based on the feature map (Wang et al., 2021b). Examples of dense prediction tasks include semantic segmentation (Zhou et al., 2017), surface normal estimation (Silberman et al., 2012), and monocular depth estimation (Eigen et al., 2014). Semantic segmentation predicts the semantic class of each pixel, surface normal estimation predicts the surface orientation of each pixel, and monocular depth estimation predicts the distance of each pixel from the camera.

**Semantic segmentation** is the task of assigning a semantic category to each pixel of an image. The fully convolution network (FCN) (Long et al., 2015) has been the fundamental work in the field of semantic segmentation. Follow up methods use atrous convolution (Chen et al., 2017a;b), pyramid pooling module (Zhao et al., 2017) or CRF/MRF (Liu et al., 2015; Liang-Chieh et al., 2015; Zheng et al., 2015) to help refine the coarse predictions by FCN. Recently, motivated by the success of Transformer architecture (Vaswani et al., 2017), SETR (Zheng et al., 2021), MaX-DeepLab (Wang et al., 2021a), MaskFormer (Cheng et al., 2021), SegFormer (Xie et al., 2021), EVA (Fang et al., 2022) have proved the effectiveness of Transformer-based architectures for semantic segmentation.

**Surface normal estimation** is to extract 3D geometry and surface orientation from a single image. Traditional approaches design networks with high capacity to learn both global and local cues. For example, Deep3D (Wang et al., 2015) proposes a two-stream CNN to fuse coarse scene structure and fine details. SkipNet (Bansal et al., 2016) introduces a skip-network approach to recover fine object details. PAP (Zhang et al., 2019) proposes pattern-affinitive propagation to utilize the matched non-local affinity information. TiltedSN (Do et al., 2020) introduces a spatial rectifier to transform the surface normal distribution and

handle tilted images. Moreover, GeoNet (Qi et al., 2018) enforces depth-normal consistency and incorporates geometric relation based on two-stream CNNs. EAU (Bae et al., 2021) introduces a new parameterization for the surface normal probability distribution to address the aleatoric uncertainty.

**Monocular depth estimation** requires estimating depth from a single RGB image. Methods based on deep convolutional neural networks have been proposed to achieve good accuracy, including the usage of radar modalities (Lo & Vandewalle, 2021), adopting multi-scale network architecture (Laina et al., 2016; Hu et al., 2019; Xu et al., 2021), using geometric constraints (Yin et al., 2021), pretaining wit masked image modeling Xie et al. (2022) or guiding by local planar (Lee et al., 2019). Recently, AdaBins (Bhat et al., 2021) leverages a Transformer-based architecture block to perform global processing of the scene's information. Depthformer (Agarwal & Arora, 2022) introduces an encoder-decoder Transformer network to predict multiscale feature maps.

However, each task requires a unique network design, resulting in significant differences among networks for different tasks. In this paper, we aim to unify the networks and expressions of various dense prediction tasks.

## 2.2 Vision Model Unification

Recently, there has been a significant amount of research in the field of computer vision Kokkinos (2017); Wang et al. (2020); Yuan et al. (2021) focused on developing models that can perform a wide range of tasks without the need for task-specific architectures. These unified models are able to achieve good results on a variety of tasks and have attracted significant attention from researchers.

Examples of these unified models include Perceiver (Jaegle et al., 2021b), Perceiver IO (Jaegle et al., 2021a) and Uni-Perceiver v2 (Li et al., 2022), which propose general architectures that can handle data from arbitrary settings. OFA (Wang et al., 2022) designs a unified sequence-to-sequence framework for a diverse set of multimodal and uni-modal understanding and generation tasks. Pix2Seq (Chen et al., 2021) proposes a generic framework for object detection using sequences of discrete tokens, and Pix2Seq v2 (Chen et al., 2022a) extends this approach to tasks such as segmentation, keypoint estimation, and image captioning. Flamingo (Alayrac et al., 2022) introduces a universal structure that produces natural language outputs for a variety of cross-modal tasks, demonstrating the effectiveness of using a unified architecture for multi-task and multimodal tasks in vision and language.

However, these models typically perform tasks that inherently generate discrete outputs or natural language, while our proposed approach, UniPixTask, focuses on dense prediction tasks that produce pixel-level predictions. Concurrent to our work, UViM (Kolesnikov et al., 2022) proposes a discrete guiding code for unifying a set of vision tasks like panoptic segmentation, depth prediction, and colorization. It relies on the use of both discrete codes and the original image to perform the task. Unified-IO (Lu et al., 2022) homogenizes different forms of input and output for a wide set of tasks into a sequence of discrete vocabulary tokens, leveraging VQ-GAN (Esser et al., 2021) to learn a label-to-label mapping and obtain the discrete tokens. In this paper, we propose UniPixTask, which is distict from them. UniPixTask models discrete representation from task-specific structures using vector quantization. It benefits from the well-established task decoder and the intermediate semantics contained in the mapping from image to label. We also incorporate modules for compressing the sequence to improve the generation efficiency of our approach.

## 3 UniPixTask

In this section, we introduce UniPixTask: a unified approach to model dense prediction tasks by discrete latent variables. It consists of three main parts: (i) a Sequence Learner to generate a unified expression of the input data in an autoregressive manner; (ii) a novel Dense Decoder to construct discrete codes for various dense prediction tasks; (iii) the unified training objective and the approach to obtain task-specific predictions.

### 3.1 Sequence Generation

The goal of Sequence Learner is to perform various tasks with discrete tokens drawn from a unified and finite vocabulary. As language is inherently discrete, inputs and outputs across a wide number of NLP tasks are represented as a sequence of symbols. Similarly, vision tasks like keypoints detection and object detection can be represented by a set of discrete points. To benefit from this homogeneous property, we construct Sequence Learner by the language modeling approach to generate discrete tokens for dense prediction tasks.

Sequence Learner is modeled by an encoder-decoder architecture. The decoder is responsible for autoregressive generation while the encoder extracts image representation. The decoder predicts one token at a time, conditioned on the image representation and the preceding sequence. Both modules are composed of stacked Transformer layers (Vaswani et al., 2017), which consist of self-attention, feed-forward neural network, layer norm and residual learning. Specially, a cross-attention module is additionally included in each layer of the decoder to perform a correlation learning between tokens and images.

### 3.2 Discrete Construction of Dense Predictions in Latent Feature Space

Different from most vision tasks, the dense prediction task which produces diverse forms of pixel-level predictions poses great challenges for unification. For example, the output of semantic segmentation is the predicted category for each pixel while the output of surface normal estimation is the three values for x/y/z orientations. Simply quantizing these outputs into discrete variables will inevitably lead to decline of accuracy and loss of local semantic details. Experimental results of this scheme can be also found in Section 4.2.

Meanwhile, since the autoregressive decoder predicts code by code, the length of predicted sequence will significantly influence the computational costs. For example, considering a 6-layer autoregressive Transformer, predicting a map with $80 \times 80$ codes needs to go through 38,400 layers. Due to the quadratically increasing cost in sequence length, the traditional model is hard to be scaled to higher resolutions.

Therefore, to address these problems, we propose Dense Decoder with compression and restoration modules to achieve efficient and high-quality transformation of discrete codes. The model is a combination of two mapping module: an encoded mapping from the image input to discrete tokens $f : \mathcal{X} \to \mathcal{V}$ and a decoded mapping from tokens to restored output $g : \mathcal{V} \to \mathcal{Y}_i$. In general, $\mathcal{X}$, $\mathcal{V}$, $\mathcal{Y}$ represent the image space, discrete space, and output space respectively. The index $i$ represents different forms of outputs in dense prediction tasks. Unlike simple quantization at the output level, we focus on the intermediate feature space in order to confront the accuracy decline under the discretization. Accordingly, the goal of Dense Decoder is to take $\mathbf{x} \in \mathcal{X}$ as the input, model the discrete representation $\mathbf{v} \in \mathcal{V}$, and finally be able to construct the output $\mathbf{y} \in \mathcal{Y}$.

As discussed above, the quantization in $\mathcal{V}$ should be a collection of finite and discrete representations. We raise the idea from vector quantization (VQ) (Van Den Oord et al., 2017), where the representations can be drawn from a learned codebook. To be specific, the vocabulary $\mathcal{V} \in \mathbb{R}^{k \times d}$ is a set of $k$ vectors $v \in \mathcal{V}$ with $d$ dimensions. Note that $\mathcal{V} = \{v_i\}_{i=1}^{k}$, $v_i \in \mathbb{R}^d$. Then Dense Decoder will learn how to choose codes from the vocabulary to perform a task with pixel-level labels.

The encoded mapping $f$ consists of a Vision Transformer (Dosovitskiy et al., 2020) backbone, a compression module, and a tokenizer. Given an input image $\mathbf{x} \in \mathbb{R}^{H \times W \times 3}$, the vision transformer first projects the input into vectors and extracts the feature $\mathbf{z} \in \mathbb{R}^{h \times w \times c}$, where $(H/h, W/w)$ is the projected patch size and $c$ is the channel dimension. Then the following compression module reduces the size of the feature map to $(h', w')$ by learnable down-sampling layers. In practice, the total number of features is compressed to a quarter of the original map, indicating that the predictions of corresponding discrete tokens are able to reduce the computational cost to 1/4.

Finally, the tokenizer projects the compressed features into code space $\mathbf{z}' \in \mathbb{R}^{h' \times w' \times d}$ (where the channel dimension $d$ is typically much smaller than $c$) and obtain the discrete tokens by mapping onto the nearest neighbor in the codebook $\mathcal{V}$:

$$\mathbf{e}_i = v_j, \text{ where } j = \mathrm{argmin}_j \|\mathbf{z}'_i - v_j\|_2. \tag{1}$$

$\mathbf{e} = f(\mathbf{x})$ is exactly the discrete representation we want. It is chosen from the finite discrete space, that is, $j \in \{1, 2, ..., k\}$. On the one hand, it is correspondingly generated from each image input. And on the other hand, it is capable to predict the task-specific output for dense prediction tasks.

Then this transformation from the discrete codes to the task-specific output space is achieved by the decoded mapping $g$. The most challenging goal of $g$ is to use compressed tokens to accomplish dense predictions with high quality. Accordingly, besides task output heads, a restoration module is proposed to combine into this decoded mapping. The module can be summarized into three parts, channel restoration, spatial restoration, and a smooth transformer.

Firstly, the channel restoration is performed by the efficient vision Transformer. It projects the channel dimension of discrete tokens $d$ to that of the latent feature space $c$. Then, a learnable up-sampling approach as spatial restoration is engaged to construct the original spatial size of feature map. On this basis, the smooth transformer will be used to obtain $\tilde{\mathbf{z}} \in \mathbb{R}^{h \times w \times c}$, making up for the inconsistency of semantics after the first two restoration. Finally, with this continuous high-dimensional feature, we can use an arbitrary existing task head to acquire the prediction $\mathbf{y}$ for the desired task. The total process of decoded mapping can be expressed by $\mathbf{y} = g(\mathbf{e})$.

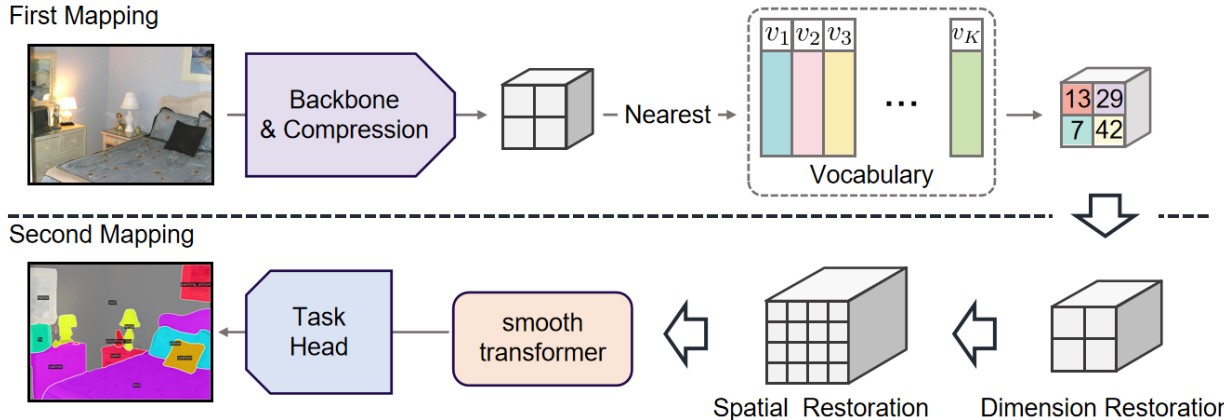

Figure 2: The structure of Dense Decoder. It consists of two mappings. The output of the encoded mapping and the input of the decoded mapping are discrete code representations, which are shown in the upper right of the figure. The encoded mapping consists of a backbone, a compression module, and a tokenizer, while the decoded mapping consists of three restoration modules and task heads.

### 3.3 Training Objectives

To obtain a well-trained Dense Decoder, the objectives can be divided into two parts. On the one hand, the learning of discrete code representations is shared by all tasks. And on the other hand, the restoration and prediction of outputs are specific for each task.

When constructing discrete representations, the key is to minimize the conversion loss when mapping the compressed features onto the codebook. Thus a commitment loss (Van Den Oord et al., 2017) is adopted. The objective can be written as:

$$\mathcal{L}_f = \|\mathrm{sg}[\mathbf{z}'] - \mathbf{e}\|_2^2 + \|\mathbf{z}' - \mathrm{sg}[\mathbf{e}]\|_2^2, \tag{2}$$

where $sg[\cdot]$ denotes the stop-gradient operation. The first term optimizes the codebook, moving the embedding vectors toward the output features in code space. Meanwhile, the second term updates the Transformer backbone, compression module, and the projection layer in the encoded mapping of Dense Decoder, constraining the output to a limited set of discrete space.

Next, the training of Dense Decoder also requires the help of task-specific objectives. They are used to optimize the decoded mapping which needs to be able to effectively restore the task output and meanwhile

help construct the code space to support the task. We employ task-specific models to generate the training labels for the restored features output by restoration modules in the decoded mapping. Given a well-trained task model $T$, it generally consists of two parts, a feature extraction backbone $T_1$ and a task output head $T_2$. The objective is to minimize the difference between task features $T_1(\mathbf{x})$ and the restored features from discrete codes:

$$\mathcal{L}_t = \|T_1(\mathbf{x}), \hat{\mathbf{z}}\|_2. \tag{3}$$

Here we use the mean squared error to supervise this restoration. Then when the features restored by the Dense Decoder are consistent with the features extracted by $T_1$, we can directly use $T_2$ as the task output head in Dense Decoder to obtain the final task predictions.

For Sequence Learner, the goal is to predict the distribution of possible indices for the next code $p(e_i|\mathbf{x}, e_{<i})$ based on the image representation and previous codes $e_{<i}$. Thus the training objective of Sequence Learner is a general loss function for all tasks, which is to maximize the log-likelihood of the predicted representations (Van Den Oord et al., 2017):

$$\mathcal{L}_e = \sum_i \hat{e}_i \log p(e_i|\mathbf{x}, e_{<i}). \tag{4}$$

$\mathbf{x}$ is the input image and $\hat{e}$ is the one-hot label generated by the code prediction of well-trained Dense Decoder. When the code predicted by Sequence Learner is consistent with that of the encoded mapping in Dense Decoder, we can then leverage the decoded mapping to perform the prediction of task output.

Therefore, at the inference stage, generating the dense prediction by a given image therefore involves two modules: Sequence Learner to obtain discrete tokens, and the decoded mapping in Dense Decoder to predict the task output. The sequence is generated code by code while each is sampling the largest likelihood predicted by Sequence Learner, *i.e.*, $p(e_i|\mathbf{x}, e_{<i})$. Then once the sequence with a certain length is generated, it can be fed into Dense Decoder for restoration and prediction of task-specific outputs.

### 3.4   Advantages over VQ-GAN.

The main difference between our method and VQ-GAN (Esser et al., 2021) in Unified-IO (Lu et al., 2022) is that VQ-GAN learns a label-to-label mapping, while our method uses vector quantization to learn an image-to-label mapping. Therefore, our UniPixTask has three advantages over VQ-GAN. First, our model can keep up with the times by improving the structure and accuracy of the model for each task, because it is able to use the task head of almost all task-specific models as its decoder. For a well-established model, we can first insert vector quantization and our proposed restoration modules between the backbone and the task head, and take its task head as our well-trained task decoder. And then after training with Equation 2 and Equation 3, we can then obtain the great discrete code representations for the task. However, it is difficult for VQ-GAN to catch up with the latest task-related models in terms of accuracy, as it can only be helped by improvements in the relevant image-to-image model and cannot be helped by task-related models.

Second, when training Dense Decoder and Sequence Learner for a task, we only need a well established model, not even the data used for training. The labels that the Dense Decoder needs to use are the hidden feature variables obtained by the well-trained model. In contrast, for VQ-GAN, a corresponding decoder needs to be trained using labels for each task dataset.

Third, for VQ-GAN in Unified-IO, to obtain a task-specific decoder head in unified vision tasks, it needs to learn a mapping from labels to labels. For example, in the segmentation task, the label needs to be converted into a segmentation map, where different colors represent different categories. Then the decoder learns the mapping from discrete codes to colors. However, this latent variable is limited to the label level, without semantic-aware image-to-label mapping, which is not an effective latent representation for auto-regressive encoder to perform the task. On contrast, UniPixTask learns the mapping from images to task labels based on the complete task process and well-established model to obtain discrete latent variables effective for the task. In general, the Sequence Learner of the UniPixTask only needs to perform the backbone part of the task-specific model, which will be beneficial for future multi-modal or multi-task unification.

# 4 Experiments

We apply UniPixTask to three pixel-level labeling vision tasks: semantic segmentation, surface normal estimation and monocular depth estimation. We describe the unified settings and select task-specific output head for each task below. Quantitative results are presented in Table 1,2,3 respectively and qualitative results are shown in Figure 4.

## 4.1 Implementation Details

**Sequence Learner** Any backbone architecture can be compatible with the encoder. In experiments, we adopt Swin-B (Liu et al., 2021) as the encoder and add a simple feature pyramid module with $3 \times 3$ convolution layers to extract multi-scale features. We then construct a 6-layer Transformer (Vaswani et al., 2017) as decoder to predict code in autoregressive way.

**Dense Decoder** For backbone model, we select Swin-B as well and also use a feature pyramid module to obtain multi-scale features. Then we use a combination of a $3 \times 3$ convolution layer with 2 strides and a hyperbolic tangent function to construct the compression module. For all the projection layers, we directly use a $1 \times 1$ convolution layer.

In the decoded mapping, the channel restoration is composed of two $3 \times 3$ convolution layers with a tangent function in the middle and a subsequent 3-layer Transformer with 768 hidden size. Then a transposed convolution is adopted as the dimension restoration. Finally, another 3-layer Transformer is employed as the smooth transformer. More details can be found in the appendix.

## 4.2 Baseline

We design a baseline model for clearer comparison, which structure is shown in Figure 3. Rather than looking for a latent variable feature space for unification, the baseline model directly quantizes the output space into a finite sequence of discrete codes. For fair comparison, we limit the length of codes for the baseline same as the codes used in UniPixTask. We first down-sample the ground-truth map and then convert it to discrete codes as new labels.

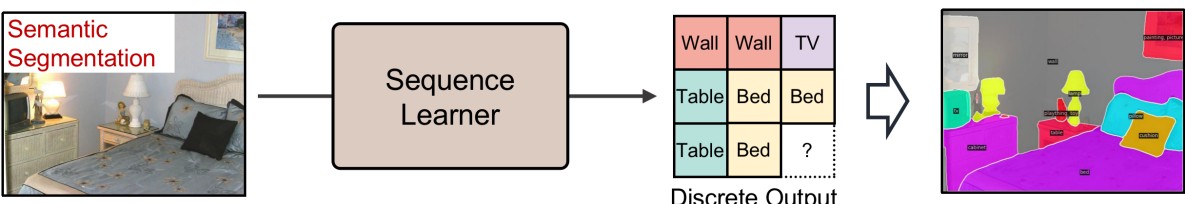

Figure 3: The structure of baseline model.

Notably, the baseline model fails to perform both surface normal and depth estimation tasks, as its result is shown in Table 2 and Table 3. The poor performances are due to the high requirements of details and resolutions on these two tasks. The results demonstrate that the outputs for some pixel-labeling tasks are difficult to directly express only by 1/256 discrete codes. UniPixTask provides an appealing solution with high accuracy based on the same amount of codes.

## 4.3 Semantic Segmentation

Semantic segmentation requires assigning each pixel of an image a semantic category. Each value of the output is a class index representing the predicted category in that pixel. We follow MaskFormer (Cheng et al., 2021) to construct the task-specific output head. The tokenizer has 1024 dictionary vectors where each has a length of 16.

We train on ADE20k (Zhou et al., 2017) dataset, which is a popular scene parsing benchmark containing dense labels of 150 categories. It includes 20K images in the training set, and 2K images in the validation set. For evaluation, we follow MaskFormer to use the standard metric mIoU (mean Intersection-over-Union) (Everingham et al., 2015). The training crop size is $640 \times 640$, while Sequence Learner will predict a length of $20 \times 20$ codes to perform the segmentation task. For the baseline, we use the index of the class label which is in discrete form and has a fixed mapping.

Table 1 shows that UniPixTask achieves 48.5 mIoU on semantic segmentation. It significantly outperforms the baseline and Unified-IO with the same backbone. And compared to the task-specific model, UniPixTask is only 4.2 points lower than MaskFormer. We analyze that the accuracy gap with the task-specific model comes from the loss of details where we only use very few tokens to express the full image. Moreover, Figure 4 demonstrates that our model is capable of producing a segmentation map of similar quality to that of the task-specific model.

## 4.4 Surface Normal Estimation

Surface normal estimation predicts three values of x/y/z orientations for each pixel. We adopt the decoder of EAU model (Bae et al., 2021) as the task output head. We combine the Swin-B model and the decoder, and then retrain it to obtain the ready task head. The tokenizer has 1024 dictionary vectors where each has a length of 64. For the baseline, since the ground-truth map of x/y/z orientations are normalized and each orientation value is in the range of $[0, 1]$, we divide the value of x and y axes into 32 bins respectively and obtain the value of z axis by the normalization formula. Therefore, the total number of codes required for the surface normal label is 1024.

We evaluate UniPixTask on NYUv2 (Silberman et al., 2012), which consists of 464 indoor scenes with RGB-D video sequences. As the official training set only contains 795 images, recent methods leverage on additional images to reconstruct the training set (Bansal et al., 2016; Qi et al., 2018; 2020). To ensure a fair comparison, we follow the same training and dataset settings as EAU (Bae et al., 2021). The training crop size is $480 \times 640$ and Sequence Learner will predict a length of $30 \times 40$ codes.

Table 2 compares the accuracy of UniPixTask and other related works. Compared to task-specific approaches, our method is only 1.7 rmse higher than EAU. Meanwhile, it outperforms other task-specific models, demonstrating the effectiveness of the proposed unified architecture.

## 4.5 Monocular Depth Estimation

Monocular Depth Estimation requires estimating the depth map where each value is an arbitrary positive number. We follow AdaBins (Bhat et al., 2021) to construct the task output head. The tokenizer has 1024 dictionary vectors where each has a length of 64. For the baseline, we divide the ground-truth depth map into 256 bins. Each code represents a specific bin that can be transferred to a depth value.

We train on NYU Depth v2 (Silberman et al., 2012), which provides images and depth maps for different indoor scenes. 24,231 image-depth pairs are divided for the training set and the rest 654 pairs are for the test set. The training crop size is $480 \times 640$, and the corresponding predicted sequence has a length of $30 \times 40$.

Table 3 shows that UniPixTask achieves 0.414 rmse on depth estimation, outperforming UViM and Unified-IO by significant margins. It is only 0.05 worse than the task-specific model AdaBins, demonstrating UniPixTask contributes an effective unified way for pixel-labeling tasks.

It is worth noticing that due to the discrete code modelling from continuous feature space, UniPixTask has a performance gap with latest state-of-the-art approaches. However, this accuracy gap is small and controllable, since our model is able to keep up with times by changing the task decoder of recent task-specific models for each task. In other words, we propose a high-quality solution for dense prediction tasks based on discrete code representations. UniPixTask actually achieves a combination of both the great performances and the generality to more modalities.

| Method | Backbone | Crop Size | mIoU (s.s.) ↑ | mIoU (m.s.) ↑ |
|---|---|---|---|---|
| DeepLabV3+ (Chen et al., 2017a) | ResNet-101 | $512 \times 512$ | 45.5 | 46.4 |
| MaskFormer (Cheng et al., 2021) | ResNet-101 | $512 \times 512$ | 46.0 | 48.1 |
| SETR (Zheng et al., 2021) | ViT-L | $512 \times 512$ | - | 50.3 |
| ViT-Adapter-B (Chen et al., 2022b) | ViT-B | $512 \times 512$ | 50.7 | 51.9 |
| Swin-UperNet Liu et al. (2021) | Swin-B | $640 \times 640$ | - | 51.6 |
| Mask2Former (Cheng et al., 2022) | Swin-B | $640 \times 640$ | 53.9 | 55.1 |
| MaskFormer (Cheng et al., 2021) | Swin-B | $640 \times 640$ | 52.7 | 53.9 |
| **Dense Decoder (Ours)** | Swin-B | $640 \times 640$ | 50.5 | 52.0 |
| Baseline | Swin-B | $640 \times 640$ | 41.4 | - |
| Unified-IO$^\dagger$ (Lu et al., 2022) | Swin-B | $640 \times 640$ | 43.2 | - |
| **UniPixTask (Ours)** | Swin-B | $640 \times 640$ | 48.7 | 50.5 |

Table 1: Comparison of the proposed method and related works for semantic segmentation on ADE20k. The upper part of the table shows task-specific approaches, while the lower part shows unified models which represent the task by unified discrete codes. $^\dagger$ denotes our reproduction of the method by replacing the backbone to Swin-B for a fair comparison. We report both single-scale (s.s.) and multi-scale (m.s.) for evaluation. Our method outperforms unified models and is competitive with recent task-specific approaches.

| Method | mean ↓ | median ↓ | rmse ↓ | 11.25° ↑ | 22.5° ↑ | 30° ↑ |
|---|---|---|---|---|---|---|
| SkipNet (Bansal et al., 2016) | 19.8 | 12.0 | 28.2 | 47.9 | 77.0 | 77.8 |
| SURGE (Wang et al., 2016) | 20.6 | 12.2 | - | 47.3 | 68.9 | 76.6 |
| GeoNet (Qi et al., 2018) | 19.0 | 11.8 | 26.9 | 48.4 | 71.5 | 79.5 |
| PAP (Zhang et al., 2019) | 18.6 | 11.7 | 25.5 | 48.8 | 72.2 | 79.8 |
| GeoNet++ (Qi et al., 2020) | 18.5 | 11.2 | 26.7 | 50.2 | 73.2 | 80.7 |
| IronDepth (Bae et al., 2022) | 20.8 | 11.3 | 31.9 | 49.7 | 70.5 | 77.9 |
| EAU (Bae et al., 2021) | 14.9 | 7.5 | 23.5 | 62.2 | 79.3 | 85.2 |
| **Dense Decoder (Ours)** | 16.7 | 9.4 | 24.7 | 55.8 | 75.7 | 82.8 |
| Baseline | 48.3 | 47.1 | 54.5 | 6.6 | 19.9 | 29.4 |
| **UniPixTask (Ours)** | 17.1 | 9.8 | 25.2 | 54.6 | 75.0 | 82.2 |

Table 2: Accuracy of surface normal estimation on NYUv2. The upper part of the table shows task-specific approaches, while the lower part shows unified models which represent the task by unified discrete codes. The baseline model has poor performance due to the high requirements of details and the constraint of 1/256 codes. Our method achieves state-of-the-art performances under the same setting.

| Method | Backbone | $\delta_1 \uparrow$ | $\delta_2 \uparrow$ | $\delta_3 \uparrow$ | REL $\downarrow$ | RMS $\downarrow$ |
|---|---|---|---|---|---|---|
| RSIDE (Hu et al., 2019) | SENet-154 | 0.866 | 0.975 | 0.993 | 0.115 | 0.530 |
| SARBN (Chen et al., 2019) | SENet | 0.878 | 0.977 | 0.994 | 0.111 | 0.514 |
| EGC (Yin et al., 2019) | ResNeXt-101 | 0.875 | 0.976 | 0.994 | 0.108 | 0.416 |
| BTS (Lee et al., 2019) | DenseNet-161 | 0.885 | 0.978 | 0.994 | 0.110 | 0.392 |
| P3Depth (Patil et al., 2022) | ResNet101 | 0.898 | 0.981 | 0.996 | 0.104 | 0.356 |
| PixelFormer (Agarwal & Arora, 2023) | Swin-L | 0.929 | 0.991 | 0.998 | 0.090 | 0.322 |
| AdaBins (Bhat et al., 2021) | EfficientNet-B5 | 0.903 | 0.984 | 0.997 | 0.103 | 0.364 |
| **Dense Decoder (Ours)** | Swin-B | 0.888 | 0.985 | 0.997 | 0.116 | 0.386 |
| Baseline | Swin-B | 0.365 | 0.627 | 0.798 | 0.374 | 1.163 |
| Unified-IO | T5-B | - | - | - | - | 0.469 |
| UViM | ViT-L | - | - | - | - | 0.467 |
| **UniPixTask (Ours)** | Swin-B | 0.864 | 0.979 | 0.995 | 0.126 | 0.414 |

Table 3: Comparison of the proposed method and related works for depth estimation on NYYU-Depth-v2 dataset. The upper part of the table shows task-specific approaches, while the lower part shows unified models which represent the task by unified discrete codes. The baseline model fails again due to the high requirements of details and the constraint of 1/256 codes. Under the same length of codes, our method outperforms Unified-IO and UViM and is competitive with recent task-specific approaches.

## 5 Key Issues and Discussion

### 5.1 Influence of Code Length and Codebook Size

We study how the length of discrete code and size of the codebook affect the performance. We vary the two dimensions of vocabulary: the number of vectors $k$ and the dimension $d$. Nine models are trained with different settings: a cross-product of $k \in \{16, 64, 256\}$ and $d \in \{256, 1024, 4096\}$, to show a clear comparison. Table 4 reports the comparison results of Dense Decoder on semantic segmentation while Table 5 reports the results of UniPixTask.

| | | Codebook Size | | |
|---|---|---|---|---|
| | | 256 | 1024 | 4096 |
| Code Length | 16 | 49.1 | 50.5 | 51.3 |
| | 64 | 48.3 | 49.2 | 49.4 |
| | 256 | 47.1 | 48.4 | 48.0 |

| | | Codebook Size | | |
|---|---|---|---|---|
| | | 256 | 1024 | 4096 |
| Code Length | 16 | 48.5 | 48.7 | 48.5 |
| | 64 | 45.2 | 47.1 | 46.7 |
| | 256 | 44.8 | 46.1 | 46.5 |

Table 4: Performances of Dense Decoder under different settings of codebook size and code length on ADE20k (measured as single-scale mIoU).

Table 5: Performances of Sequence Learner under different settings of codebook size and code length on ADE20k (measured as single-scale mIoU).

The experiments are held on ADE20k for semantic segmentation. For Dense Decoder, when the code length is relatively small, *i.e.*, 16 and 64, the accuracy of the model benefits from the increase of codebook size. However, when the code length reaches 256, the use of the tokenizer in Dense Decoder will be saturated quickly, resulting in a decrease of accuracy after 1024 codes. Meanwhile, when the codebook size is fixed, the results show that the accuracy of Dense Decoder benefits from a shorter code length.

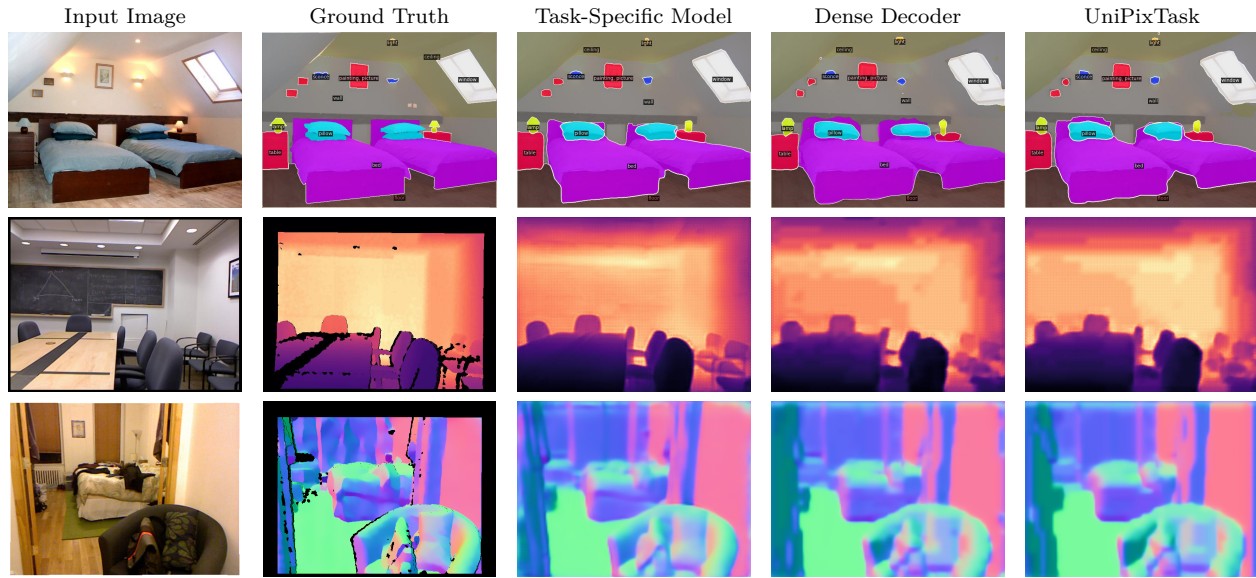

| Input Image | Ground Truth | Task-Specific Model | Dense Decoder | UniPixTask |

Figure 4: Visualizations of our model on three pixel-labeling tasks: semantic segmentation (1st row), monocular depth estimation (2nd row) and surface normal estimation (3rd row). The columns represent the visualization results predicted by different models. Dense Decoder quantizing by discrete codes is able to generate near perfect results. And the UniPixTask predicting in autoregressive way also achieves very high quality, demonstrating its effectiveness.

For Sequence Learner, it is rather unclear how the accuracy is affected by size of codebook and the length of codes. In general, when the codebook size is fixed, the model is still benefiting from a shorter code length. However, for different code lengths, all optimal sizes of codebooks in Dense Decoder have changed. We analyze that longer codes and larger codebook will have potentially negative effect on autoregressive-predicted models.

## 5.2 Utilization rate of Codebook

Next, we further investigate the codebook utilization rate under the variation of code length and codebook size to to explore potential reasons for the performance fluctuations. For the Dense Decoder, we count how many codes in the vocabulary have been used at least once during the inference stage. Intuitively, a short length of code will have limited ability to express semantic information and thus require more vocabulary to achieve the task. Figure 5 reports the utilization rates under the same cross-product of nine settings in Section 5.1.

As expected, with the same size of the codebook, the utilization rate of shorter codes in the codebook is higher than that of longer codes. Meanwhile, for codebooks with the same code length, the number of used codes will increase accordingly with the expansion of the vocabulary size. For example, for a codebook with a code length of 16, when the optional vocabulary is only 256, the number of used codes is 252. However, when the optional vocabu-

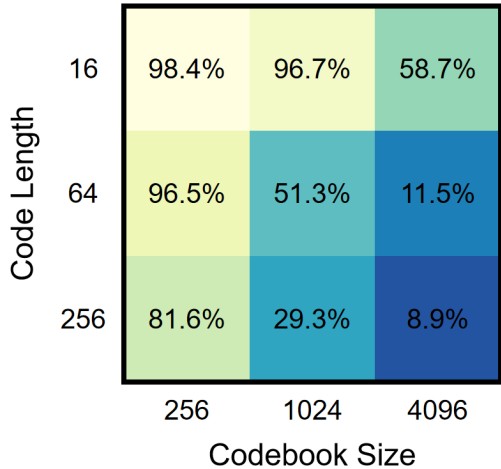

Figure 5: The utilization rate of the codebook in Dense Decoder. The experiments are held on ADE20k for the semantic segmentation task under different settings. The utilization rates are measured on validation set.

lary is increased to 1024, the number of used codes will be increased to 990. This shows that the increase in the optional vocabulary will encourage the model to spread the task semantics to more codes and avoid overloading a few codes. However, for a codebook with a relatively low utilization rate, such as a codebook with a code length of 256 and a size of 1024, increasing its vocabulary size hardly improves the total usage. At this time, a large number of unused codes in the codebook are redundant.

## 5.3 Computational Cost Evaluation

We evaluate the memory usage, model size, and inference time of four models, including baseline, Unified-IO reproduced by Swin-B, Dense Decoder, and the final UniPixTask. The result is shown in Table 6. We also take the original Unified-IO with T5-B backbone for comparison. We compute the floating point operations (FLOPs) and report fps (images/s) on the input with $640 \times 640$ crop size. The experiments are held on Tesla V100 32G. Since Dense Decoder predicts the whole image in only one forward pass rather than in an autoregressive way, its flops and fps are significantly better than other methods. It is worth noting that despite adopting a task-specific decoder and leveraging on restoration modules in Dense Decoder, UniPixTask still achieves lower memory usage and similar fps compared to reproduced Unified-IO under the same backbone. It thus shows that the proposed components are not only efficient but also lightweight.

| Method | Backbone | flops (G) | params (M) | fps (images/s) |
|---|---|---|---|---|
| Baseline | Swin-B | 1152.07 | 94.7 | 0.69 |
| Unified-IO | T5-B | - | 241.0 | - |
| Unified-IO$^\dagger$ | Swin-B | 1366.5 | 135.4 | 0.28 |
| MaskFormer | Swin-B | 139.7 | 88.8 | 12.14 |
| Dense Decoder* | Swin-B | 183.3 | 126.3 | 10.48 |
| UniPixTask | Swin-B | 1195.5 | 137.8 | 0.28 |

Table 6: Computational cost evaluation. $^\dagger$ denotes our reproduction of the method with Swin-B backbone. * denotes that the method does not predict in autoregressive way. Despite adopting a task-specific decoder and leveraging on restoration modules in Dense Decoder, UniPixTask still achieves lower memory usage and similar fps compared to reproduced Unified-IO under the same backbone.

We also hold experiments to study the computational cost of each component in UniPixTask and compare with the cost of VQ-GAN decoder in Unified-IO. SL denotes Sequence Learner; EM denotes Encoded Mapping; and DM denotes Decoded Mapping. The result is shown in Table 7. For the decoder part, the EM module is very lightweight and has very little impact on the overall computation and parameter cost of the model. And the computational cost is concentrated on the DM module, which consists of multiple transformer layers to perform a spatial restoration and a smooth module. However, the combination of decoder head in UniPixTask still achieves lower floating point operations than that of VG-GAN decoder in Unified-IO.

| Backbone | SL | EM | DM | Task Head | VQ-GAN Decoder | flops (G) | params (M) |
|---|---|---|---|---|---|---|---|
| ✓ | | | | | | 138.9 | 82.6 |
| ✓ | ✓ | | | | | 1148.1 | 94.6 |
| ✓ | ✓ | ✓ | | | | 1148.1 | 94.6 |
| ✓ | ✓ | ✓ | ✓ | | | 1190.7 | 131.5 |
| ✓ | ✓ | ✓ | ✓ | ✓ | | 1195.5 | 137.8 |
| ✓ | | | | | ✓ | 1366.5 | 135.4 |
| | | ✓ | ✓ | ✓ | | 43.4 | 43.2 |
| | | | | | ✓ | 214.5 | 40.8 |

Table 7: Computational cost evaluation of different components. The computational cost of UniPixTask decoder is concentrated on the DM module, which consists of multiple transformer layers to perform restoration and smooth modules.

# 6 Conclusion

In this paper, we introduce UniPixTask, a unified approach that achieves state-of-the-art performance in dense prediction tasks. The key idea behind UniPixTask is to unify the representations of various tasks using discrete codes. It consists of two components: Sequence Learner, which learns to generate predictions in an autoregressive manner, and Dense Decoder, which models diverse inputs and outputs using discrete codes with task semantics. We demonstrate that our model performs effectively on semantic segmentation, surface normal estimation, and monocular depth estimation tasks. In the future, we plan to extend UniPixTask to support additional tasks and modalities.

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

# A  Appendix

We provide detailed architectures of UniPixTask (Table **??**) and the settings of training hyperparameters of three pixel-labeling tasks: semantic segmentation (Table 9), surface normal estimation (Table 10) and monocular depth estimation (Table 11).

## A.1  Network Architecture

## A.2  Hyperparameters for Semantic Segmentation Training

| Input | Layer | Output | Output Dimension |
|---|---|---|---|
| image | - | - | $H \times W \times 3$ |
| **Encoded Mapping** | | | |
| image | Swin-B | $\mathbf{z}$ | $H/16 \times W/16 \times 512$ |
| $\mathbf{z}$ | $\begin{pmatrix} \text{Conv2D(C}_{out}\text{=512, ks=3, stride=2, padding=1),} \\ \text{Tanh(),} \\ \text{Conv2D(C}_{out}\text{=64, ks=1, stride=1, padding=1)} \end{pmatrix}$ | $\mathbf{z}'$ | $H/32 \times W/32 \times 64$ |
| $\mathbf{z}'$ | tokenizer in Eq. 1 | $\mathbf{e}$ | $H/32 \times W/32 \times 64$ |
| **Decoded Mapping** | | | |
| $\mathbf{e}$ | 3-layer transformer | $\tilde{\mathbf{z}}_1$ | $H/32 \times W/32 \times 512$ |
| $\tilde{\mathbf{z}}_1$ | $\begin{pmatrix} \text{ConvTranspose2D(C}_{out}\text{=512, ks=3, stride=2, padding=1),} \\ \text{GroupNorm()} \end{pmatrix}$ | $\tilde{\mathbf{z}}_2$ | $H/16 \times W/16 \times 512$ |
| $\tilde{\mathbf{z}}_2$ | 3-layer transformer | $\tilde{\mathbf{z}}_3$ | $H/16 \times W/16 \times 512$ |
| $\tilde{\mathbf{z}}_3$ | $\begin{pmatrix} \text{Conv2D(C}_{out}\text{=512, ks=3, stride=2, padding=1),} \\ \text{Tanh(),} \\ \text{Conv2D(C}_{out}\text{=64, ks=1, stride=1, padding=1)} \end{pmatrix}$ | $\tilde{\mathbf{z}}$ | $H/16 \times W/16 \times 512$ |
| $\tilde{\mathbf{z}}$ | Task Head | Task Output | - |

Table 8: Detailed network architecture of UniPixTask.

| Hyperparameters | Values |
|---|---|
| Autoregressive Layers | 6 |
| Restored Channel | 256 |
| Hidden Size | 512 |
| FFN Hidden Size | 2048 |
| Attention Heads | 8 |
| Dropout | 0.1 |
| Pre-Norm | ✓ |
| Patch Size | $16 \times 16$ |
| Codebook Size | $1024 \times 16$ |
| Input Size | $640 \times 640$ |
| Batch size | 16 |
| Adam | ✓ |
| Peak Learning Rate | 0.00006 |
| Minimal Learning Rate | 1e-6 |
| Warmup Epochs | 1500 |
| Gradient clipping | $\times$ |
| Weight decay | 0.01 |
| Random Flip | ✓ |
| Random Crop | ✓ |

Table 9: Hyperparameters for Semantic Segmentation on ADE20k.

| Hyperparameters | Values |
|---|---|
| Autoregressive Layers | 6 |
| Restored Channel | 256 |
| Hidden Size | 512 |
| FFN Hidden Size | 2048 |
| Attention Heads | 8 |
| Dropout | 0.1 |
| Pre-Norm | ✓ |
| Patch Size | $8 \times 8$ |
| Codebook Size | $1024 \times 64$ |
| Input Size | $480 \times 640$ |
| Batch size | 16 |
| Adam | ✓ |
| Learning Rate | 0.00036 |
| Gradient clipping | 0.1 |
| Weight decay | 0.01 |
| Random Flip | ✓ |
| Color Augmentation | ✓ |
| Random Crop | ✓ |

Table 10: Hyperparameters for Surface Normal Estimation on NYUv2.

| Hyperparameters | Values |
|---|---|
| Autoregressive Layers | 6 |
| Restored Channel | 256 |
| Hidden Size | 512 |
| FFN Hidden Size | 2048 |
| Attention Heads | 8 |
| Dropout | 0.1 |
| Pre-Norm | ✓ |
| Patch Size | $8 \times 8$ |
| Codebook Size | $1024 \times 64$ |
| Input Size | $480 \times 640$ |
| Batch size | 16 |
| Adam | ✓ |
| Learning Rate | 0.0006 |
| Gradient clipping | $\times$ |
| Weight decay | 0.01 |
| Random Flip | ✓ |
| Random Rotation | ✓ |
| Random Crop | ✓ |

Table 11: Hyperparameters for Monucular Depth Estimation on NYU Depth v2.

