# OpenReview forum: "Unifying Pixel-Labeling Vision Tasks by Sequence Modeling"
_TMLR — Rejected by TMLR_

### Review · Reviewer_P5Nb · 2023-02-13

**Summary Of Contributions:**

The paper proposes a two-step modeling approach for dense vision tasks.

The first step is to train a deep model that maps an input image to the dense output mask. This model is called “dense decoder” and has a special feature: it applies a vector quantization procedure to the intermediate feature map using the methodology proposed in the [VQVAE paper](https://arxiv.org/abs/1711.00937). The part of the model before the quantization step is denoted as “encoder” $f$, and the part after the quantization step is denoted as “decoder” $g$.

Then, as the second step, the paper proposes to train an autoregressive transformer-based model (“sequence learner”) to model the discrete sequence output of $f$. At test time “sequence learner” predicts the discrete sequence and then “decoder” $g$ maps the discrete code to the task output.

The proposed model is evaluated on the three tasks: semantic segmentation, surface normal and depth estimation.

Unfortunately, I think that the proposed model does not contribute to the field: it does not carry any new insight and is not useful in practice (see the explanation in the next section).


**Audience:**

No

**Claims And Evidence:**

No

**Requested Changes:**

The paper appears to have flawed motivation, is a special case of the existing method (UViM) and also fails to credit prior research appropriately. Unfortunately, I do not have any suggestions that will make this paper publishable at TMLR.

**Strengths And Weaknesses:**

Weaknesses:

(1) The paper fails to appropriately cite the prior work. Large fraction of the paper’s technical content is based on the VQVAE paper, which is not even cited. More concretely, equation 1 in the paper is essentially the equation 2 of the VQVAE paper, and equation 2 of the paper is the equation 3 in the VQVAE paper. The paper should make a clear relation to the VQVAE paper.

From the model unification angle, the highly related work, namely Unified-IO and UViM, is mentioned, but it is argued that “These works leverage VQ-GAN (Esser et al., 2021) to encode pixel-level labels as discrete tokens. However, this approach is limited by the structure and conversion precision of VQ-GAN.” It is not clear why VQ-GAN would be more limiting than VQVAE and, crucially, UViM does not even use VQ-GAN. It also uses VQVAE, the same approach as the paper uses.

Finally, the paper's model is the special case of the UViM model, where “encoder” $f$ does not have access to the ground truth data. I argue below that such a special case is not a good idea.

(2) I think the motivation behind the proposed approach is flawed. As the paper demonstrates, the “dense decoder” itself is capable of solving all the tasks considered in the paper. If needed, unification can be simply done via using separate output heads. So there is no need in “representation quantization” and an additional “sequence learner” model.

The baselines with the heavy manual quantization lead to bad results, as the majority of ground-truth signal is lost. I do not think they support the proposed model.

I would like to stress that the proposed approach would most likely fall apart for any task that requires high-level output coordination beyond just having strong pixel-level predictions. For example it will fail for the task of panoptic segmentation. The reason is that the “dense encoder” is a simple feed-forward model with pixel-level loss and is not equipped with the ability to coordinate structured outputs (or intermediate codes that encode these outputs). Papers like Unified-IO and UViM, which also tackle model unification, address this problem by letting the “encoder” $f$ to have access to the ground-truth label, while the proposed model is lacking access to ground-truth when producing the discrete code.

---

> ### Author Response · Authors · 2023-03-13
> **Response to Reviewer P5Nb**
>
> Thank you very much for your great effort in reviewing our paper.
>
> **Q1**: Difference from UViM.
>
> **A1**: There are three obvious differences between our model and UViM. First, the decoder in our model does not need to use the original image to complete the task, which is relatively shallow, making the model more efficient and lightweight for future multitask integration. Second, when training dense decoders and sequence learners for a task, we only need a well established model, not even the data used for training. The labels that the dense decoder needs to use are the hidden feature variables obtained by the well-trained model. Third, our model can keep up with the times by improving the structure and accuracy of the model for each task, because it is able to use the task head of almost all task-specific models as its decoder. For a well-established model, we can first insert vector quantization and our proposed restoration modules between the backbone and the task head. We take its task head as our well-trained decoder. And then after training with Eq.2 and Eq.3, we can then obtain the great discrete code representations for the task.
> These differences allow our model to significantly outperform UViM for depth estimation. Thank you for your concern. We will discuss in detail about the related works.
>
> **Q2**: The rationality behind ‘representation quantization’ and ‘sequence learner’.
>
> **A2**: The purpose of these two structures is to use discrete codes to perform multi-task and multi-modal unification within a single architecture. Language is a sequence of symbols that are discrete in nature. Therefore, the easiest way to unify vision and language in the short term will be to use classification to predict discrete codes. In this way, the code can be converted directly to the word, and a shallow task decoder will be adopted for most vision tasks. In contrast, if continuous latent representations are used, additional decoders will be required to make final predictions for both text and visual images.

---

### Review · Reviewer_EJNN · 2023-02-21

**Summary Of Contributions:**

This paper presents a sequence modeling method to unify vision tasks including surface normal estimation, monocular depth estimation and semantic segmentation. Different from closely related work unified-io, this paper proposes to use a more complex latent code to model the relations between input and output. The method achieves competitive performance on three tasks based on Swin-B.

**Audience:**

Yes

**Claims And Evidence:**

Yes

**Requested Changes:**

- The paper can be stronger if better results are provided and more powerful baseline are compared.

- I suggest the authors to add more details about architectures and training process to help readers to better understand the models. Code for reproduce results in the paper would be better.

**Strengths And Weaknesses:**

Strengths:

- The paper overall is easy to follow.

- The method can achieve better results than Unified-IO reproduced by the authors. The method also largely outperforms the baseline which suggests the basic design is effective.

Weaknesses:

- The technical contribution of this paper is relatively low. Adding a channel dimension to the discrete code used in Unified-IO is close to previous practices like multi-scale encoding used in VG-GAN (Esser et al., 2021).  The overall framework is very close to Unified-IO.

- The model is only compared with some reproduced baselines. The proposed method generally underperforms some widely used methods like MaskFormer. Some more advanced methods like Mask2Former are not compared.

- Some important details are missing. For example, the detailed architectures for semantic/spatial/channel restoration are not mentioned.

---

> ### Author Response · Authors · 2023-03-13
> **Response to Reviewer EJNN**
>
> Thank you very much for your great effort in reviewing our paper.
>
> **Q1**: Detailed architectures.
>
> **A1**: Detailed architectures for semantic/spatial/channel restoration can be found in Section 4.1. We also add more details in the appendix. And our code will be available.
>
> **Q2**: The difference between our method and Unified-IO.
>
> **A2**: The main difference between our method and Unified-IO is that Unified-IO uses VQ-GAN to learn a label-to-label mapping, while our method uses vector quantization to learn an image-to-label mapping. Therefore, our UniTask has three advantages over VQ-GAN.
> First, our model can keep up with the times by improving the structure and accuracy of the model for each task, because it is able to use the task head of almost all task-specific models as its decoder. For a well-established model, we can first insert vector quantization and our proposed restoration modules between the backbone and the task head. We take its task head as our well-trained decoder. And then after training with Eq.2 and Eq.3, we can then obtain the great discrete code representations for the task. However, it is difficult for VQ GAN to catch up with the latest task-related models in terms of accuracy, as it can only be helped by improvements in the relevant image-to-image model and cannot be helped by task-related models.
>
> Second, when training dense decoders and sequence learners for a task, we only need a well established model, not even the data used for training. The labels that the dense decoder needs to use are the hidden feature variables obtained by the well-trained model. In contrast, for VQ-GAN, a corresponding decoder needs to be trained using labels for each task dataset.
>
> Third, for VQ-GAN in Unified-IO, to obtain a task-specific decoder head in unified vision tasks, it needs to learn a mapping from labels to labels. For example, in the segmentation task, the label needs to be converted into a segmentation map, where different colors represent different categories. Then the decoder learns the mapping from discrete codes to colors. However, this latent variable is limited to the label level, without semantic-aware image-to-label mapping, which is not an effective latent representation for auto-regressive encoder to perform the task. On contrast, UnitTask learns the mapping from images to task labels based on the complete task process and well-established model to obtain discrete latent variables effective for the task. In general, the sequence learner of the UniTask only needs to perform the backbone part of the task-specific model, which will be beneficial for future multi-modal or multi-task unification.
>
> **Q3**: More state-of-the-art methods and performance comparison.
>
> **A3**: We have updated recent SoTA methods for each task. It is worth noticing that due to the discrete code modelling from continuous feature space, our model has a performance gap with latest state-of-the-art approaches. However, this accuracy gap is small and controllable, since our model is able to keep up with times by changing the task decoder of recent task-specific models for each task. In other words, we propose a high-quality solution for dense prediction tasks based on discrete code representations. UniTask actually achieves a combination of both the great performances and the generality to more modalities.

---

### Review · Reviewer_DJEx · 2023-03-01

**Summary Of Contributions:**

This work proposes a unified architecture, UniTask, for three pixel-wise tasks: semantic segmentation, surface normal estimation, and monocular depth estimation.
The same encoder, latent coding, and decoder modules are applied to each task.
The encoding backbone is taken from prior work, but the encoding of features into a latent discrete code differs from contemporary work on unified vision models (UViM and Unified-IO) that make use of VQ-GAN coding.
The decoder is designed to restore channel and spatial dimensions of the output from the latent codes, and then further process the representation to improve accuracy (the so-called "semantic" restoration step).
Results with the proposed encoder/auto-regressive discrete coding/decoder model improve on concurrent unified architectures but lag behind task-specific models.
The main empirical contributions of this work are the comparison with other unified architectures and the demonstration that a discrete code model can work for this purpose without the VQ-GAN component.
The technical contributions are less immediately apparent, given that the encoding architecture (Swin-B) and the coding objective (related to VQ-VAE) are taken from prior work, and the decoder makes use of common architectural components for pixel-wise decoding.
That said, the particular design choices made, and the exact proposed decodeer architecture, seem fairly effective across the three tasks studied, at least for these two datasets.



**Audience:**

Yes

**Broader Impact Concerns:**

None. This work is on the general topic of pixel-wise tasks in vision and it does not present any specific ethical issues in itself.


**Claims And Evidence:**

No

**Requested Changes:**

**Critical Changes for Acceptance**

Please consider the claims w.r.t. the state-of-the-art and provide further detail about the computation and ablation of the proposed model.

- Revise the results claim to acknowledge the state-of-the-art is higher on the given benchmarks. The results in this paper are valid, in that they control for backbone (sometimes), but they do not rival the state-of-the-art.
- Report a more detailed measurement of the computation required by the various components of the model.
- Ablate the stages of the dense decoder, to understand the importance of each type of restoration.
- Discuss related work and credit prior work where it is made use of, such as the commitment loss of the VQ-VAE (to highlight a particular example).

**Minor Changes for Improvement**

Please consider this feedback on the exposition to make the paper more accessible and informative to a broader audience.

- The experiments (Sec. 4) could use a forward pointer to the analysis and discussion in Sec. 5. Many readers will want to know that the key factors like the codebook size and code length are examined, and may first look for this in the experiments section.
- The qualitative results in Fig. 4 would be more informative if good (cherry-picked) and bad (lemon-picked) results were shown for the highest and lowest scoring inputs of the validation set. For precision, each result could be shown alongside its score to connect the qualitative result to the quantitative result. As a further step, it would be nice to see more qualitative results in the supplement, which the main text could then point to.

**Miscellaneous Feedback**

- "UniTask" is a quite broad name, and something more precise like "UniPixTask" might be more appropriate, given the focus on pixel-labeling tasks.
- The wording in the figures in the text could be more specific:
  - Fig. 2 "first" vs. "second" mapping are generic names, which could be replaced with encoding and decoding.
  - "semantic restoration" is a vague and unscientific term, and should be replaced by a description of the operation as attention or of the purpose as context processing or the like.
- COCO [Lin et al. 2014] is a strange reference for semantic segmentation. PASCAL VOC is an earlier and landmark benchmark, or ADE20K is a currently popular benchmark (as studied in this submission), so consider changing the citation.



**Strengths And Weaknesses:**

**Strengths**

This work examines a timely topic, experiments with several tasks (as is necessary for work on unified modeling across tasks), and shows results that improve on concurrent methods (but not always with strong controls for comparability).

- How to unify vision tasks is a current topic, and how to unify pixel-wise tasks deserves special attention due to the spatial precision and computational scale required.
- The experiments cover a diverse choice of pixel-wise tasks: semantic segmentation, surface normals, and depth. These include a recognition/semantic task (semantic segmentation) and reconstruction/physical tasks (normals, depth). The tasks span two datasets, ADE20K and NYUDv2.
- The accuracies of the proposed UniTask improve on the accuracies of other unified methods, specifically UViM and Unified-IO. However, note that they do not always control for backbone (depth estimation in Table 3), and as such are not comparable experiments.
- While related to other unified architectures in its use of discrete codes, this work does differ by departing from VQ-GAN, and so in this way indicates an alternative approach to quantization for these types of models.
- Sec 5. analyzes the quantization/coding dimensions of the proposed method for code length, codebook size, and code utilization rate and reports the computational cost of the proposed UniTask, the internal baseline for this work, and the existing Unified-IO method.

**Weaknesses**

This work requires attention to improve its experimental results, discussion of prior work, and clarity.

The experimental results for UniTask are not competitive w.r.t. task-specific results and the proposed baselines are not convincing.

- The task baselines are not up-to-date and not the state-of-the-art.
  - semantic segmentation: [EVA (arXiv'22)](https://arxiv.org/abs/2211.07636), [ViT-Adapter (ICLR'23, arXiv'22)](https://arxiv.org/abs/2205.08534)
  - surface normals: [IronDepth (BMVC'22)](https://arxiv.org/abs/2210.03676)
  - depth: [PixelFormer (WACV'23, arXiv)](https://arxiv.org/abs/2210.09071v1), [SwinV2-L 1K-MIM (arXiv)](https://arxiv.org/abs/2205.13543v2)
- Each task is limited to a single dataset. To measure how unified the proposed architecture is, it would be more convincing to show that it works for more than one instance of each task.
  For example, semantic segmentation could also be evaluated on Cityscapes, and depth and normals could be evaluated on KITTI (although other choices are also possible and acceptable).
- Limited analysis and ablation: while Sec. 5 takes on the key issues of code design and computation it does not sufficiently study them for the purposes of this work.
  - The quantization analysis in Tables 4 & 5 is only carried out for semantic segmentation. As the subject of this work is the unification of pixel-wise tasks, it is necessary to do this analysis for the three tasks.
  - The baseline provided by this work, with the sequential encoder and direct discretization of the output, controls for the size of codes but does not control for the amount of computation and choice of autoregressive encoding. Per Table 6 the computational cost of UniTask is 10x the computation of a non-autoregressive model, and so it is reasonable to wonder how a non-autogressive baseline that controls for computation would perform. That is, the decoder-less baseline could have a longer/denser code or a larger/longer codebook, and still be more computationally efficient than the proposed full model. As an alternative baseline, a Swin-B encoder _without_ discrete codes could have a longer/denser latent representation coupled with a shallow decoder for each task would be reasonable.
  - The encoding and decoding architectures are taken as fixed and neither the sequence learner or dense decoder architectures are ablated. The decoder has several stages, so knowing how much each contributes would guide further development of decoders by future work.
- The results for each task have a specific tuning of the dictionary size and code length. If this depends on the task, dataset size, or both is not empirically investigated. However, if a different tuning is required for each task, then this weakens the claim of task unification.
- The analysis suggests that UniTask may scale poorly to larger codes ("We analyze that longer codes and larger codebook predicted models" on pg. 10), although this may be a function of dataset size. Many architectures, such as the Swin-B backbone used in this work, exhibit results that improve with larger dimensions (such as their channel widths or layer depths).

Related work is missing, and it is closely related to the purpose of this work in unifying visual tasks or discretely coding representations.

  - Unified modeling for pixel-wise tasks: [UberNet (CVPR'17)](https://arxiv.org/abs/1609.02132) designs not only a unified architecture but a unified, multi-task model for a variety of pixel-labeling tasks at once that include the set considered in this submission.
  - Single architectures for dense prediction: [HRNet PAMI'20](https://arxiv.org/abs/1908.07919) or [HRFormer NeurIPS'21](https://arxiv.org/abs/2110.09408) (if a focus on attention is desirable) both define a single architecture that handles a variety of dense tasks (detection, segmentation, keypoints) with simple decoders.
  - Unified architecture for vision and language tasks: [Uni-Perceiver v2 (arXiv)](https://arxiv.org/abs/2211.09808). This work includes several vision and language tasks, although this submission specializes in pixel-wise tasks, among which Uni-Perceiver v2 only looks at semantic segmentation.
  - The commitment loss in particular and the training in general (Sec. 3.3) needs to indicate novelty and prior work as the case may be. For instance, a commitment loss is part of the [VQ-VAE](https://arxiv.org/abs/1711.00937) but it is presented here without reference to prior work.

There are issues of clarity with the overall interface for the model, that is how it is applied across tasks, the method details, and the self-containment of the work.

- UniTask is a single architecture for multiple tasks, but it is not a multi-task model. That is, one learned set of parameters does not do the three tasks experimented with in this work. This could be spelled out early on, in the abstract or the introduction, and especially in Fig. 1 where the three tasks are summarized.
- The training of the sequence learner/encoder component of the model is ambiguous. Is the objective shared across all tasks (Eq. 4) but the model is trained separately for each task---as understood for the purpose of this review---or are the encoder parameters shared while the decoder parameters are separate?
- "However, this approach is limited by the structure and conversion precision of VQ-GAN. In this paper, we propose an alternative approach that models discrete representation in the intermediate feature space, allowing us to benefit from arbitrary well-established structures." What are these structures?
- The task-specific output heads are not specified in their details, at least not in the text, and only identified by reference. The same goes for the Transformer blocks included in the encoder and decoder. These could at least be reproduced in the supplement for reference without pulling up the relevant papers.

**Summary**

This work proposes a particular architecture, UniTask, for unifying pixel-wise tasks through discrete latent codes.
The variety of tasks selected is satisfactory, in that one is semantic (semantic segmentation) and two are physical (normals and depth), and that they cover two datasets (ADE20K, NYUDv2).
Comparing UniTask with the concurrent work of UViM and Unified-IO shows accuracy improvements (at least on the subset of tasks selected across the union of the experiments in these papers).
The results for all of the unified models do not yet compete with state-of-the-art specific models, but there is progress.
At present the issues for acceptance are (1) the disagreement between the claims w.r.t. the task-specific models and (2) the lack of thorough analysis of UniTask (either in ablation of its own components or close comparison with its concurrent works).
This work would be more informative if it could explain how its design results in the different results w.r.t. UViM and Unified-IO rather than simply report that they differ.

**Questions**

- How sensitive is each task to the tuning of the discrete coding? Is there a scheme for choosing the dictionary size and code length automatically? This would make the approach more unifying, if less model search were needed per task.
- How much does each step of decoding (channel/spatial/semantic restoration) contribute?
- What is the computational cost of each component of UniTask? More precisely, how much is needed for the backbone, encoding, and decoding? Table 6 is a good start, but does not report how much of the cost is due to auto-regressive encoding vs. backbone encoding, for instance, which is what would make UniTask differ from alternatives.
- This is more pragmatic than conceptual, but what is the motivation for relying on discrete codes? Would a model be less unified if it were to rely on continuous latents and discrete or continuous decoders? Attaching a shallow, task-specific head on a shared representation is not so much work to require for each task.

---

> ### Author Response · Authors · 2023-03-13
> **Response to Reviewer DJEx**
>
> Thank you very much for your great effort in reviewing our paper.
>
> **Q1**: More task-specific state-of-the-art methods.
>
> **A1**: We have updated recent SoTA methods for each task and acknowledged the performance gap. The gap is due to the discrete code modelling from continuous feature space. However, this accuracy gap is small and controllable, since our model is able to keep up with times by changing the task decoder of recent task-specific models for each task. In other words, we propose a high-quality solution for dense prediction tasks based on discrete code representations. UniTask actually achieves a combination of both the great performances and the generality to more modalities.
>
> **Q2**: Why rely on discrete latent instead of continuous one.
>
> **A2**: Our research focuses on designing a single architecture for multiple tasks, including multiple modalities. Language is a sequence of symbols that are discrete in nature. Therefore, the easiest way to unify vision and language in the short term will be to use classification to predict discrete codes. In this way, the code can be converted directly to the word, and a shallow task decoder will be adopted for most vision tasks. In contrast, if continuous latent representations are used, additional decoders will be required to make final predictions for both text and visual images.
>
> **Q3**: Specific tuning of the dictionary size and code length for each task.
>
> **A3**: Thank you for your concern. The default setting is 64/1024 for the dictionary size and code length, which achieves best results for depth estimation and surface normal estimation. On semantic segmentation, it also achieves satisfactory performance, which is only a worse of 1.6 mIoU than the best setting 16/1024. Since our goal and future direction focus on the unified output of language and images, we believe that with this common setting of 64&1024, the model can achieve satisfactory results for most vision tasks.
>
> **Q4**: The training objective in Eq. 4.
>
> **A4**:The ‘shared’ loss function can be replaced by the ‘general’ loss function. Since the dense decoder is closely related to the task, it requires training with task-specific loss. In contrast, the sequence learner predicts in an auto-regressive way and performs classification for discrete codes. Therefore, regardless of the type of task, the same loss function, log-likelihood in Eq. 4, can be used for the sequence learner. Notice that the learning goal is different (generated by the task-specific dense decoder).
>
> **Q5**: Arbitrary well-established structures.
>
> **A5**: The main advantage of UniTask is that it is able to use the task head of almost all task-specific models as its decoder. For a well-established model, we can first insert vector quantization and our proposed restoration modules between the backbone and the task head. We take its task head as our well-trained decoder. And then after training with Eq.2 and Eq.3, we can then obtain the great discrete code representations for the task.
>
> **Q6**: The difference between our method and VQ-GAN.
>
> **A6**: As discussed in Q5, our method can apply to different well-established task-specific models. It gives UniTask two advantages over VQ-GAN.
> First, our model can keep up with the times by improving the structure and accuracy of the model for each task, thus improving performance. However, it is difficult for VQ GAN to catch up with the latest task-related models in terms of accuracy, as it can only be helped by improvements in the relevant image-to-image model and cannot be helped by task-related models. Second, when training dense decoders and sequence learners for a task, we only need a well established model, not even the data used for training. The labels that the dense decoder needs to use are the hidden feature variables obtained by the well-trained model. In contrast, for vqgan, a corresponding decoder needs to be trained using labels for each task dataset. Third, for VQ-GAN, it learns a mapping from images to same images. To obtain a task-specific decoder head in unified vision tasks, such as Unified IO, VQ-GAN needs to learn the discrete latent variables that map labels to labels. However, this latent variable is limited to the label level, without any semantic-aware image-to-label mapping, which is not an effective latent representation for auto-regressive encoder to perform the task. On contrast, UnitTask learns the mapping from images to task labels based on the complete task process and well-established model to obtain discrete latent variables effective for the task.
>
> **Q7**: ‘UniPixTask’ and ‘UniTask’; encoding and decoding; semantic restoration; COCO citation; Detailed task-specific structures, more ablation studies about computational cost.
>
> **A7**: Thanks for your constructive advice. We have already revised them.
>
> **Q8**: Related works.
>
> **A8**: Thanks for the valuable references. We have updated them in the paper.

---

### Decision · Action_Editors · 2023-04-07

**Recommendation:** Reject

**Comment:**

This paper proposes an approach to perform many dense prediction tasks by first converting an input image to a sequence of discrete codes that does not depend upon the task, and then using a task-specific head to generate outputs based on this code. Although reviewers generally applauded the paper’s goal of unifying vision tasks, they raised concerns regarding the baselines, discussion of previous work, lack of ablations, lack of evidence for the claim that newer, better-performing task-specific heads can be used with the model, and lack of motivation for the use of discrete codes. In response, the authors addressed concerns regarding baselines by adding numbers obtained with recent SOTA approaches to the tables in the paper, and also expanded the discussion of related work. Although these additions strengthened the work, reviewers remained unconvinced regarding the remaining points, and all recommend or lean toward rejection. I concur with the reviewers’ recommendations.

**Audience:**

The submission probably exceeds TMLR's bar. On one hand, as reviewers noted, the discretization of the latent codes appears to serve no purpose in the proposed architecture and reduces accuracy in practice, and thus the approach appears to have little immediate significance. On the other hand, it is conceivable that future applications could benefit from discrete latent codes, as the authors indicate.

**Claims And Evidence:**

Although the performance of the proposed model appears to be properly evaluated, some claims remain unsupported. The submission claims that the framework is compatible with any combination of backbone and task head and could be adapted to SOTA models, but it shows results with only a single choice of encoder and decoder. Design choices are not fully ablated and reviewers were unconvinced by the claim that discretization of the model's hidden representation leads to practical advantages.